# Impact of negative tuberculin skin test on growth among disadvantaged Bangladeshi children

S. M. Abdul Gaffar*, Mohammod Jobayer Chisti, Mustafa Mahfuz, Tahmeed Ahmed

Nutrition and Clinical Services Division (NCSD), International Centre for Diarrhoeal Disease Research, Bangladesh (icddr,b), Dhaka, Bangladesh

* gaffar@icddrb.org

## Abstract

Millions of children are suffering from tuberculosis (TB) worldwide and often end-up with fatal outcome especially in resource-poor settings. Tuberculin skin test (TST) is a conventionally used diagnostic test, less sensitive but highly specific for the diagnosis of clinical TB especially in undernourished children. However, we do not have any data on the role of TST positivity among the children who received nutritional intervention. Our aim was to examine the growth differences between TST-positive and TST-negative undernourished children aged 12 to 18 months who received nutritional intervention prospectively for 90 feeding days. Our further aim was to explore the determinants of TST positivity at enrollment. TB screening as one of the secondary causes of malnutrition was performed on 243 stunted [length for age Z score (LAZ) <-2 standard deviations] or at-risk of stunting (LAZ score between <-1 and -2 standard deviations) children in a community-based intervention study designed to improve their growth parameters. Differences of growth between TST-positives ($n = 29$) and TST-negatives ($n = 214$) were compared using paired samples t-test and multivariable linear regression from anthropometric data collected before and after nutritional intervention. Multivariable logistic regression was used to find out possible predictors of TST positivity using baseline sociodemographic data. Of the 243 children screened, 29 (11.9%) were TST-positive and 11 (4.5%) had clinically diagnosed pulmonary TB. Statistically significant improvement of LAZ and weight for age Z-score (WAZ) were observed among the TST-negative participants at the end of intervention period ($p = 0.03$ for LAZ and $p = 0.01$ for WAZ). However, we did not find any association between TST status and response to nutritional intervention in our multivariable linear regression models. Our study findings demonstrated a positive impact of nutritional intervention on growth parameters among TST-negative participants.

## Introduction

Tuberculosis (TB) mainly affects the lungs with grave consequences unless treated appropriately in time. It is one of the leading causes of under-5 child mortality worldwide particularly

**Data Availability Statement:** All relevant data are within the manuscript and its Supporting Information files.

**Funding:** Tahmeed Ahmed received research funding from the Bill and Melinda Gates Foundation

under its Global Health Program (Project investment id is OPP1136751). The funding agency had no role in study design, data collection and interpretation, or the decision to submit the manuscript for publication. https://www.gatesfoundation.org/How-We-Work/Quick-Links/Grants-Database/Grants/2015/11/OPP1136751.

**Competing interests:** The authors have declared that no competing interests exist.

in the WHO Southeast Asia and Africa regions [1]. Apart from mortality, a good number of children are at risk of developing TB each year in the 22 high-burden countries according to the prediction of a mathematical modelling study [2]. In 2017, an estimated one million children developed TB disease worldwide [3]. Bangladesh is one of the TB endemic countries where co-existence of malnutrition and tuberculosis is rampant [2, 4, 5]. The prevalence of *Mycobacterium tuberculosis* (MTB) infection in Bangladesh according to the tuberculin survey, 2007–2009 using a cut-off point of ≥8 mm was found to be 12.4% and 22.6% among children aged 5–9 years and 10–14 years, respectively [6].

Stunting or chronic malnutrition predisposes to TB as a result of reduced host defense due to the impairment of cell mediated immunity [7]. Data related to the outcome of MTB infection and disease on growth of young children is meager. To the best of our knowledge, no study findings have been reported comparing the growth pattern between tuberculin skin test (TST)-positive and TST-negative children. Moreover, the risk factors for TST positivity are mostly unexplored in the early stages of child growth and development.

The primary aim of our study was to assess the differences of linear and ponderal growth between TST-positive and TST-negative children from a disadvantaged urban community who underwent a nutrition intervention program and were screened for secondary causes of malnutrition. Moreover, we wanted to predict determinants of TST positivity from demographic and socioeconomic data collected at the time of enrollment of each participant.

## Methods

The participants screened for TB was originally recruited from a larger community-based study titled "The Bangladesh Environmental Enteric Dysfunction (BEED)" study. Children of 12–18 months' who were stunted (length-for-age Z score [LAZ] <-2 standard deviations) or at risk of stunting (LAZ <-1 to -2 standard deviations) met eligibility criterion for enrollment in the BEED study. The objectives of BEED study include development of a histological scoring system for the diagnosis of Environmental Enteric Dysfunction (EED), identifying novel non-invasive biomarkers of EED, measurement of correlation between EED score and anthropometric attainment, determining the pathogenetic role of gut microbiota in development of EED, and targeting possible biological pathways of EED for the future therapeutic interventions. Our objective of growth difference measurement between TST-positive and TST-negative participants will complement our study objectives by determining the role of chronic subclinical infection in children's growth parameters. As the objective of our analysis is different from the main BEED study, we have calculated the sample size required for this potential analysis. Sample size was calculated using the formula of a descriptive study: $n = Z^2 p(1-p)/d^2$ and found at least 163 participants are required for this type of screening program [where, $n$ = required sample size, $Z$ = standard normal variant value for 95% confidence level (1.96), $p$ = expected proportion of the condition (12.4% according to a tuberculin survey conducted in Bangladesh), and $d$ = precision (0.05)]. However, we screened 243 children and we believe the available sample will allow us to perform a case-control analysis between TST-positive and TST-negative participants, nested in the main BEED study.

EED and latent tuberculosis infection (LTBI) are both chronic infections apparently responsible for pediatric undernutrition. On site dietary supplementation was provided with one boiled egg and 150 mL of whole milk daily for 90 feeding days which added an additional 178 kilocalories, 11.1 g protein, and 11.5 g of fat to the daily diet of the participants. Participants also received one sachet of micronutrient sprinkles containing vitamins A and C, folic acid, iron, and zinc for 60 days. Moreover, every child received one dose of anthelmintic at the onset of nutritional intervention. In addition, treatment for intercurrent illness during

intervention period was ensured from the Study Clinic. Several nutrition centers were established in the community to provide supplementation among the participants under direct supervision of the research assistants. The study protocol was approved by the Research Review Committee (RRC) and Ethical Review Committee (ERC) of International Centre for Diarrhoeal Disease Research, Bangladesh (icddr,b). More details in methodology and area of the study has been described elsewhere [8, 9].

The participants who failed to graduate to an expected LAZ level in spite of nutritional intervention were screened for TB and other diseases to find out secondary causes of malnutrition. Failure to graduate is defined as LAZ persisting <-2 standard deviations for stunted and <-1 standard deviation for at risk of stunting groups at the end of intervention. Screened children underwent a chest radiograph (CXR), TST, and physical examination for TB diagnosis. History of recent close contact (within past 12 months) and suggestive symptoms of TB were elicited through a careful interview of the mother. TST was administered using the Mantoux method to each of the children using tuberculin PPD 10 tuberculin units (TU)/0.1 mL (ARK-RAY Healthcare Pvt. Ltd., Sachin, Gujarat, India) and the readings were recorded at 48–72 hours post administration in millimeter (mm). The diagnosis of TST positivity was confirmed through measuring a skin induration of ≥10 mm in diameter [10, 11]. TST is a highly specific screening and diagnostic tool to identify latent MTB infection and disease [12–14]. The principle of tuberculin test is based on delayed type of hypersensitivity reaction resulting from stimulation of already sensitized T-cells of a person infected with MTB or other atypical mycobacteria. The entire screening procedure with subsequent management of the participants is depicted in Fig 1.

Participants anthropometry (weight and length) was measured with scales (Seca GmbH & Co. KG., Hamburg, Germany) at pre- and post-intervention following standard operating procedures. During enrollment, data on mother's age, education, and occupation along with household's average monthly income and people/room used for sleeping were collected using a pre-tested questionnaire. Maternal occupation was dichotomized into housewife and the rest category. Average monthly household income was converted from Bangladeshi taka to USD (84.50 BDT = 1 USD) and analyzed as a continuous variable. Maternal age, education, and people/room for sleeping in the household were analyzed as a continuous variable in logistic regression.

The impact of TST positivity on growth was computed using paired samples t-test and multivariable linear regression. In paired t-test, mean differences of LAZ, WAZ, and WLZ between pre- and post-nutritional intervention were calculated to assess the impact of TST status on growth. Linear regression was used to construct a model to explore the independent effect of TST positivity on our outcome variables (Δlength-for-age Z score [LAZ], Δweight-for-age Z score [WAZ], and Δweight-for-length Z score [WLZ]). Covariates considered were sex (male or female) and post-intervention age. Outcome variables were calculated subtracting pre-intervention from post-intervention LAZ, WAZ, and WLZ. Covariates were first analyzed through univariate regression against our outcome variables. However, both the covariates along with TST outcome (positive or negative) were included in multivariable models irrespective of statistical significance and this was done due to their biological plausibility aiming to have an impact on growth parameters. The method was iterated for the three independent regression models for each of our three outcome variables (ΔLAZ, ΔWAZ, and ΔWLZ). All statistical analyses were computed using STATA version 15.1 (StataCorp LP, College Station, Texas, USA).

Multivariable logistic regression was employed to predict TST positivity at enrollment. The purpose behind this analysis was to identify sociodemographic factors of TST positivity. Eight covariates were sorted out for logistic regression to predict the presence of TST positivity at

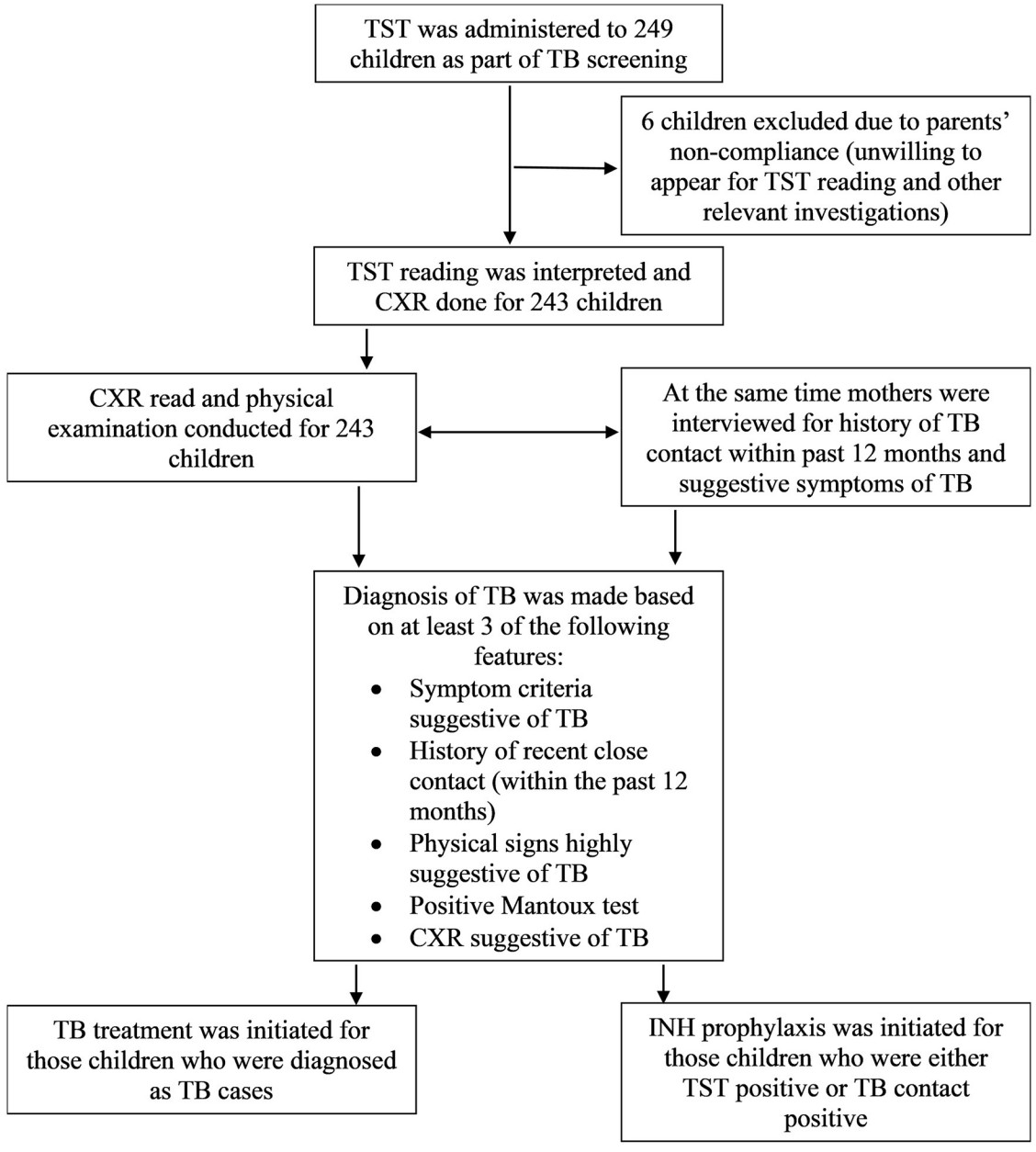

**Fig 1. Outline of TB screening study.**

enrollment based on their relevance. Individual covariate was analyzed first to detect an association with TST positivity by means of univariate regression. Significant covariates which had a *p*-value of ≤0.25 were subsequently included in multivariable analysis.

## Results

Two hundred and forty-three participants' data were included in the analysis who completed nutritional intervention and had TB screened. The sociodemographic features at enrollment are outlined in Table 1. The prevalence of TST positivity in our study was 11.9% (29/243). In stunted and at risk of stunting groups, the prevalence was 10.9% (13/119) and 12.9% (16/124),

**Table 1. Enrollment characteristics of the study participants based on tuberculin skin test (*n* = 243).**

| Parameter | TST-negative (*n* = 214) | TST-positive (*n* = 29) | *p*-value |
|:---:|:---:|:---:|:---:|
| Male | 99 (46)[a] | 16 (55) | 0.36[x] |
| At risk of stunting | 108 (50) | 16 (55) | 0.63[x] |
| Age in months | 15 (13–17)[b] | 15 (12–16) | 0.34[z] |
| Weight in kg | 8.0 ± 0.9[c] | 8.2 ± 0.9 | 0.35[y] |
| Length in cm | 72.2 ± 3.0 | 72.0 ± 3.4 | 0.71[y] |
| LAZ | -2.22 ± 0.86 | -2.23 ± 0.78 | 0.95[y] |
| WAZ | -1.86 ± 0.88 | -1.67 ± 0.74 | 0.27[y] |
| WLZ | -1.05 ± 0.88 | -0.78 ± 0.94 | 0.12[y] |
| Maternal age in years | 25 (21–29) | 24 (20–26) | 0.04[z] |
| Maternal education | 5 (1–7) | 6 (5–8) | 0.01[z] |
| Mother as housewife | 154 (72) | 26 (90) | 0.04[x] |
| Income in taka[d] | 12000 (10000–16000) | 12000 (10000–20000) | 0.91[z] |
| People/room in home | 3 (3–4) | 3 (3–4) | 0.91[z] |

[a] Count; percentage in parentheses (all such values)

[b] Median; interquartile range in parentheses (all such values)

[c] Mean ± standard deviation (all such values)

[d] One United States dollar = 84.50 Bangladeshi taka

[x] Pearson chi-square test

[y] Two-sample t test with equal variances

[z] Two-sample Wilcoxon rank-sum test

respectively [Fig 2]. TST-positive cases among male and female were 13.9% (16/115) and 10.2% (13/128), respectively [Fig 2]. Out of 29 TST-positive children, 10 were clinically diagnosed as pulmonary tuberculosis (PTB) following our national guidelines for the management of TB in children [10]. One child fulfilled the diagnostic criteria of PTB without TST positivity. Among the 11 children who were diagnosed as PTB, 6 were from at risk of stunting group and remaining children were from the stunted group. However, more female [7] than male [4] were found to have clinically diagnosed PTB.

In primary outcome measurement, we observed statistically significant LAZ and WAZ improvement among the TST-negative participants after completion of the nutritional intervention in paired samples t-test (t [213] = 2.1839, p = 0.03 for LAZ and t [213] = 2.4122, p = 0.01 for WAZ). However, in a subsample analysis, TST-positive participants without PTB showed statistically significant WAZ and WLZ improvement at the end of the intervention period in paired samples t-test (t [18] = 3.3487, p = 0.003 for WAZ and t [18] = 2.6062, p = 0.01 for WLZ). In linear regression, we did not find any statistically significant difference in LAZ, WAZ, and WLZ attainment during the intervention period between TST-positive and TST-negative participants. Moreover, TST positivity remained comparable for growth parameters (TST-positive coefficient [95% CI], *p*-value for ΔLAZ score −0.04 [−0.15, 0.08], 0.51; ΔWAZ score 0.04 [−0.11, 0.18], 0.62; and ΔWLZ score 0.09 [−0.11, 0.29], 0.37) after adjustment of age and sex of the participants in multivariable analyses. The details of the paired samples t-tests and multivariable linear regression analyses are enumerated in Tables 2–4.

In a secondary analysis to predict TST positivity, we included participants age (*p* = 0.19), maternal age (*p* = 0.03), maternal education (*p* = 0.04), and maternal occupation (*p* = 0.05) [for other than housewife; reference housewife] in our final multivariable model. However, none of the covariates achieved statistical significance for the prediction of TST positivity in

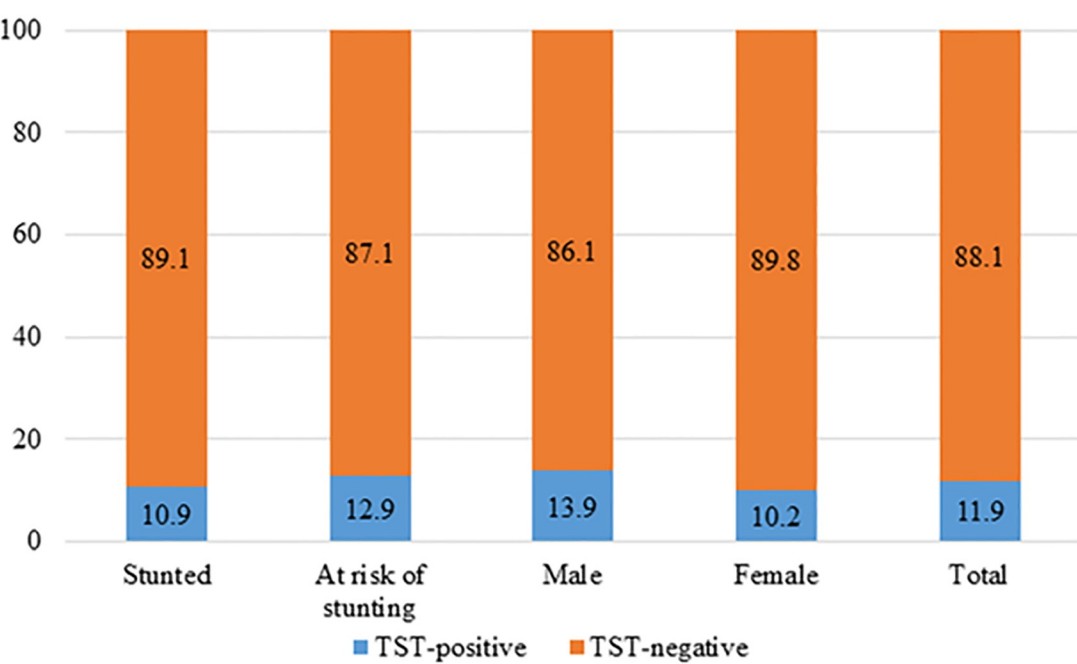

**Fig 2. Bar plots demonstrate the percentage of TST-positive and TST-negative patients stratified by enrollment group and gender.**

**Table 2. Analysis of growth differences between TST-positive and TST-negative participants using paired samples t-test.**

| Parameter | TST-positive ($n = 29$) | | | | TST-negative ($n = 214$) | | | |
|---|---|---|---|---|---|---|---|---|
| | Pre-intervention | Post-intervention | Mean Differences (Post-Pre) | *p*-value | Pre-intervention | Post-intervention | Mean Differences (Post-Pre) | *p*-value |
| **LAZ** | -2.23 ± 0.78 | -2.23 ± 0.82 | 0.006 ± 0.28 | 0.90 | -2.22 ± 0.86 | -2.18 ± 0.80 | 0.04 ± 0.30 | 0.03 |
| **WAZ** | -1.67 ± 0.74 | -1.58 ± 0.87 | 0.09 ± 0.42 | 0.26 | -1.86 ± 0.88 | -1.79 ± 0.85 | 0.06 ± 0.37 | 0.01 |
| **WLZ** | -0.78 ± 0.94 | -0.68 ± 0.95 | 0.09 ± 0.59 | 0.39 | -1.05 ± 0.88 | -1.01 ± 0.89 | 0.03 ± 0.50 | 0.30 |

Data are mean ± standard deviation

**Table 3. Analysis of growth differences between TST-positive and clinical TB participants using paired samples t-test.**

| Parameter | TST-positive ($n = 19$) | | | | Clinical TB ($n = 10$) | | | |
|---|---|---|---|---|---|---|---|---|
| | Pre-intervention | Post-intervention | Mean Differences (Post-Pre) | *p*-value | Pre-intervention | Post-intervention | Mean Differences (Post-Pre) | *p*-value |
| **LAZ** | -2.13 ± 0.75 | -2.06 ± 0.83 | 0.07 ± 0.29 | 0.32 | -2.43 ± 0.85 | -2.53 ± 0.73 | -0.11 ± 0.25 | 0.21 |
| **WAZ** | -1.54 ± 0.80 | -1.30 ± 0.88 | 0.24 ± 0.31 | 0.003 | -1.91 ± 0.57 | -2.11 ± 0.57 | -0.20 ± 0.46 | 0.21 |
| **WLZ** | -0.70 ± 1.05 | -0.44 ± 1.03 | 0.26 ± 0.44 | 0.01 | -0.93 ± 0.72 | -1.15 ± 0.55 | -0.22 ± 0.72 | 0.35 |

Data are mean ± standard deviation

**Table 4. Relationship between tuberculin skin test and anthropometric attainment of children who underwent nutritional intervention.**

| Variables | | Unadjusted Coefficient[1] (95% CI[3]) | p-value | Adjusted Coefficient[2] (95% CI) | p-value |
|---|---|---|---|---|---|
| ΔLAZ | TST (+) | -0.04 (-0.15, 0.08) | 0.51 | -0.04 (-0.15, 0.08) | 0.51 |
| | Female | -0.02 (-0.10, 0.05) | 0.54 | -0.02 (-0.10, 0.05) | 0.52 |
| | Age | 0.003 (-0.01, 0.02) | 0.70 | 0.003 (-0.01, 0.02) | 0.74 |
| ΔWAZ | TST (+) | 0.03 (-0.12, 0.17) | 0.70 | 0.04 (-0.11, 0.18) | 0.62 |
| | Female | 0.01 (-0.08, 0.10) | 0.82 | 0.01 (-0.08, 0.11) | 0.80 |
| | Age | 0.01 (-0.01, 0.03) | 0.25 | 0.01 (-0.01, 0.04) | 0.23 |
| ΔWLZ | TST (+) | 0.06 (-0.14, 0.26) | 0.55 | 0.09 (-0.11, 0.29) | 0.37 |
| | Female | 0.05 (-0.08, 0.18) | 0.47 | 0.05 (-0.08, 0.18) | 0.43 |
| | Age | 0.05 (0.02, 0.08) | 0.003 | 0.05 (0.02, 0.08) | 0.002 |

ΔLAZ, ΔWAZ, and ΔWLZ denote change in length-for-age, weight-for-age, and weight-for-length Z scores during the intervention period; TST (+) denotes tuberculin skin test positive; Age indicates post-intervention age in months

[1] Univariate linear regression

[2] Multivariable linear regression

[3] Confidence interval

**Table 5. Predictors of TST positivity at enrollment (n = 243).**

| Variables | Unadjusted OR[1] (95% CI[2]) | [3]p-value | Adjusted OR (95% CI) | [4]p-value |
|---|---|---|---|---|
| **Age of the participants** | | | | |
| (days) | 0.99 (0.99, 1.00) | 0.19 | 1.00 (0.99, 1.00) | 0.24 |
| **Age of the mothers** | | | | |
| (years) | 0.91 (0.83, 0.99) | 0.03 | 0.92 (0.84, 1.01) | 0.08 |
| **Education of the mothers** | | | | |
| (years) | 1.12 (1.00, 1.24) | 0.04 | 1.08 (0.96, 1.22) | 0.18 |
| **Occupation of the mothers** | | | | |
| **Housewife** | Reference | | Reference | |
| **Others** | 0.30 (0.09, 1.03) | 0.05 | 0.38 (0.11, 1.33) | 0.12 |

[1] Odds ratio

[2] Confidence interval

[3] Univariate logistic regression

[4] Multivariable logistic regression

multivariable logistic regression model. The details of the logistic regression analyses are listed in Table 5.

## Discussion

To our knowledge this is the first study which intended to estimate an impact of TST positivity on children's growth in response to a community-based nutritional intervention. We observed similar growth trajectories between TST-positive and TST-negative participants in response to our intervention. However, statistically significant LAZ and WAZ improvement in TST-negative children was indicative of the positive impact of nutritional intervention among them.

TST-positivity is generally indicative of MTB infection with or without active TB disease. Although, numerous studies have reported an association between malnutrition and tuberculosis [7, 15, 16], it is difficult to ascertain whether malnutrition causes tuberculosis or vice versa. Malnutrition has a profound adverse impact on the immune system of our body which exhibits a crucial role against various pathogenic microbes. Moreover, malnutrition impairs cellular host defense mechanisms that protect from bacterial infection such as MTB [7]. On the other hand, tuberculosis decreases appetite of the affected child with an increase in metabolism leading to weight loss and subsequent malnutrition [15]. However, till date, studies failed to establish direct relationship between LTBI and malnutrition [7]. Our study identified the knowledge gap between the relationship of TST positivity and stunting to some extent. Improved growth velocity in absence of TST positivity and/or active PTB disease below the age of two years is a novel finding which may imply immense public health implications. Moreover, we were unable to ascertain household contact of most of the TST-positive and clinical PTB cases in our study. This observation indicates that a considerable number of undiagnosed adult TB cases may be still available in the community [17, 18]. Previous studies conducted in developing countries revealed a noteworthy proportion of children suffering from LTBI with or without identifiable smear-positive household pulmonary TB contact [6, 19, 20]. A case-control study in Guinea-Bissau reported a prevalence of 41% TST-positive patients among household TB exposed and 22% among non-exposed yielding a ratio of 1.48 (95% CI 1.37, 1.60) between them [19].

Previously numerous studies had documented detrimental impact of acute and chronic gastrointestinal infections on linear growth in the developing countries [21–23]. However, no impact of fever and respiratory infections on growth was identified [22]. Growth faltering of children has been detected in absence of obvious clinical manifestations of diarrhea but with the presence of enteropathogens in stool [24, 25]. A multi-country longitudinal cohort study conducted in developing countries including Bangladesh found significant growth faltering in respect of LAZ attainment among children aged two years with higher enteropathogen burden in non-diarrhoeal stool [24]. The same study also revealed lower consumption of energy and lack of protein rich diet intake was responsible for growth deficits in respect of both LAZ and WAZ attainment. Another case-control study conducted in Bangladeshi malnourished children less than two years of age suggests that enteropathogens had a significant effect on their growth [25]. Although nutritional intervention had a significant positive impact on the LAZ and WAZ outcome in the TST-negative children, the same intervention alone without definitive treatment failed to have sufficient impact for the TST-positive children. It is apparently evident from the study that treatment of children having LTBI with INH prophylaxis along with proper nutritional support is essential to reduce stunting in such children.

The distinctive strength of our study was its prospective design to assess the response of nutritional treatment among undernourished TST-positive and TST-negative children. We ascertained the TST status only after the intervention period during a routine screening of secondary causes of malnutrition. Moreover, a true intervention was ensured through directly observed therapy.

Our study had a few important limitations. First, we only enrolled stunted and at risk of stunting children which had limited the generalizability of our findings. Second, we did not assign a control group without nutritional intervention to ascertain the impact of the intervention on TST status. Finally, although our sample size was adequate for a descriptive study, it did not provide adequate power to our primary outcome analysis particularly in respect to the number of TST-positive cases.

## Conclusion

TST positivity was common among our study population who were stunted or at risk of stunting living in an underprivileged urban community setting. Our study findings on the basis of paired samples t-test noted a positive impact of nutritional intervention on growth parameters among TST-negative participants, although, we did not find this impact in linear regression. However, the preliminary observation underscores the importance of identifying and promptly treating source cases. This may help to reduce the exposure of children to TB source cases, thus may reduce TB and increase TST-negativity. Moreover, the observation re-emphasizes the importance of providing INH prophylaxis in all LTBI cases recommended by the WHO in order to reduce TB disease in children.

## Supporting information

**S1 Dataset.**
(CSV)

## Acknowledgments

The authors gratefully acknowledge the contribution of field workers and participants with their family members for conduct of the study. International Center for Diarrheal Disease Research, Bangladesh (icddr,b) also gratefully acknowledges the following donors which provide unrestricted support: Government of the People's Republic of Bangladesh; Global Affairs Canada (GAC), Canada; Swedish International Development Cooperation Agency (Sida); and the Department for International Development (UKAid).

## Author Contributions

**Conceptualization:** S. M. Abdul Gaffar, Tahmeed Ahmed.

**Data curation:** S. M. Abdul Gaffar.

**Formal analysis:** S. M. Abdul Gaffar.

**Funding acquisition:** Tahmeed Ahmed.

**Investigation:** S. M. Abdul Gaffar, Mustafa Mahfuz, Tahmeed Ahmed.

**Methodology:** S. M. Abdul Gaffar, Mustafa Mahfuz, Tahmeed Ahmed.

**Project administration:** S. M. Abdul Gaffar, Mustafa Mahfuz, Tahmeed Ahmed.

**Resources:** S. M. Abdul Gaffar, Mustafa Mahfuz, Tahmeed Ahmed.

**Software:** S. M. Abdul Gaffar.

**Supervision:** S. M. Abdul Gaffar, Mohammod Jobayer Chisti, Mustafa Mahfuz, Tahmeed Ahmed.

**Validation:** S. M. Abdul Gaffar, Mohammod Jobayer Chisti, Tahmeed Ahmed.

**Visualization:** S. M. Abdul Gaffar, Mohammod Jobayer Chisti, Tahmeed Ahmed.

**Writing – original draft:** S. M. Abdul Gaffar.

**Writing – review & editing:** S. M. Abdul Gaffar, Mohammod Jobayer Chisti, Mustafa Mahfuz, Tahmeed Ahmed.

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
