## [Decision Letter · Decision Letter 0]

4 Sep 2019

PONE-D-19-21500

Impact of negative tuberculin skin test on growth among disadvantaged Bangladeshi children

PLOS ONE

Dear Dr. Gaffar,

Thank you for submitting your manuscript to PLOS ONE. After careful consideration, we feel that it has merit but does not fully meet PLOS ONE’s publication criteria as it currently stands. Therefore, we invite you to submit a revised version of the manuscript that addresses the points raised during the review process.

We would appreciate receiving your revised manuscript by Oct 19 2019 11:59PM. To enhance the reproducibility of your results, we recommend that if applicable you deposit your laboratory protocols in protocols.io, where a protocol can be assigned its own identifier (DOI) such that it can be cited independently in the future. For instructions see: http://journals.plos.org/plosone/s/submission-guidelines#loc-laboratory-protocols

We look forward to receiving your revised manuscript.

Kind regards,

HASNAIN SEYED EHTESHAM

Academic Editor

PLOS ONE

Journal Requirements:

2. Thank you for your ethics statement:

'Ethical approvals were obtained from Research Review Committee (RRC) and Ethical Review Committee (ERC) of icddr,b (protocol no: PR-16007; Version 1.03; March 1, 2016). Each participant enrolled in BEED study is treated according to what is morally right and proper. Separate consent forms are being used for children and adult participants. After complete disclosure, a signed informed consent statement is obtained from each subject. For minors, informed consent is obtained from the parents or authorised legal guardian of the subject. The consenting process takes place, preferably, in the residence of the subject.".   

a.Please amend your current ethics statement to include the full name of the ethics committee/institutional review board(s) that approved your specific study.

b.Once you have amended this/these statement(s) in the Methods section of the manuscript, please add the same text to the “Ethics Statement” field of the submission form (via “Edit Submission”).

Additional Editor Comments (if provided):

Major Revision

Reviewers' comments:

Reviewer's Responses to Questions

**Comments to the Author**

1. Is the manuscript technically sound, and do the data support the conclusions?

Reviewer #1: Yes

Reviewer #2: No

2. Has the statistical analysis been performed appropriately and rigorously? 

Reviewer #1: Yes

Reviewer #2: Yes

3. Have the authors made all data underlying the findings in their manuscript fully available?

Reviewer #1: Yes

Reviewer #2: No

4. Is the manuscript presented in an intelligible fashion and written in standard English?

Reviewer #1: Yes

Reviewer #2: Yes

5. Review Comments to the Author

Reviewer #1: This study has aimed to explore the determinants of TST positivity in children. TB screening as one of the secondary causes of malnutrition was performed on 243 stunted or at-risk of stunting children. Screened children underwent a chest radiograph (CXR), TST, and physical examination for TB diagnosis.

The current study has examined the growth differences between TST-positive and TST-negative undernourished children aged 12 to 18 months who received nutritional intervention prospectively for 90 feeding days.

This is the first study which intended to estimate an impact of TST positivity on children’s growth in response to a community-based nutritional intervention.

Although the current study was designed towards understanding the tuberculosis in malnutrition children, unexpectedly, nothing significant found in the study.

This study should be published in a medical journal for reporting the relationship between tuberculosis and malnutrition in children, but the study is not fit for an international peer-reviewed journal like PLOS ONE.

Reviewer #2: The manuscript presented no new significant findings. Moreover, it is already known fact to use INH prophylaxis in paediatric populations. The control group lacks significant number of cases to justify and establish the hypothesis raised. The study also document several limitations, raising questions on study design and case controlling. Outcome is not justified significantly. It is encouraged to increase the study participants and include more objectives rather than only nutritional aspects in paediatric populations.

6. PLOS authors have the option to publish the peer review history of their article (what does this mean?). If published, this will include your full peer review and any attached files.

Reviewer #1: No

Reviewer #2: No

---

## [Author Response · Author response to Decision Letter 0]

8 Oct 2019

Reviewers’ comments to PLOS ONE questions and to the author with author responses:

Question 1: Is the manuscript technically sound, and do the data support the conclusions?

Reviewer #1: Yes

Response: Thank you.

Reviewer #2: No

Response: Thank you. We have now revised the conclusion in the text for better clarity and hope revised conclusion will qualify for the publication.

Question 2: Has the statistical analysis been performed appropriately and rigorously?

Reviewer #1: Yes

Response: Thank you.

Reviewer #2: Yes

Response: Thank you.

Question 3: Have the authors made all data underlying the findings in their manuscript fully available?

Reviewer #1: Yes

Response: Thank you.

Reviewer #2: No

Response: Thank you. The full dataset was uploaded during the manuscript submission process. 

Question 4: Is the manuscript presented in an intelligible fashion and written in standard English?

Reviewer #1: Yes

Response: Thank you.

Reviewer #2: Yes

Response: Thank you.

5. Review Comments to the Author

Reviewer #1: This study has aimed to explore the determinants of TST positivity in children. TB screening as one of the secondary causes of malnutrition was performed on 243 stunted or at-risk of stunting children. Screened children underwent a chest radiograph (CXR), TST, and physical examination for TB diagnosis.

The current study has examined the growth differences between TST-positive and TST-negative undernourished children aged 12 to 18 months who received nutritional intervention prospectively for 90 feeding days.

This is the first study which intended to estimate an impact of TST positivity on children’s growth in response to a community-based nutritional intervention.

Although the current study was designed towards understanding the tuberculosis in malnutrition children, unexpectedly, nothing significant found in the study.

This study should be published in a medical journal for reporting the relationship between tuberculosis and malnutrition in children, but the study is not fit for an international peer-reviewed journal like PLOS ONE.

Response: Thanks for your overall positive comments. Our primary objective was to examine the growth differences between TST-positive and TST-negative undernourished children aged 12 to 18 months who received nutritional intervention prospectively for 90 feeding days in the parent Bangladesh Environmental Enteric Dysfunction (BEED) study. We observed statistically significant improvement of length for age Z-score (LAZ) and weight for age Z-score (WAZ) among the TST-negative participants at the end of intervention period. Our secondary objective was to explore the determinants of TST positivity at enrollment. We observed several factors (maternal age, education, and occupation) that were significantly associated with TST positivity in our study children during univariate logistic regression analysis. However, as you mentioned, multivariable logistic regression failed to identify any association of those factors with TST positivity. As you rightly noticed this is the first study which is intended to estimate an impact of TST positivity on children’s growth in response to a community-based nutritional intervention, and importantly, the objectives of the study are aligned with the methods, results, and conclusion, we feel the revised manuscript bears merit for publication in the PLOS ONE. 

Reviewer #2: The manuscript presented no new significant findings. Moreover, it is already known fact to use INH prophylaxis in paediatric populations. The control group lacks significant number of cases to justify and establish the hypothesis raised. The study also documents several limitations, raising questions on study design and case controlling. Outcome is not justified significantly. It is encouraged to increase the study participants and include more objectives rather than only nutritional aspects in paediatric populations.

Response: Thank you. Improved growth velocity in absence of TST positivity and/or active PTB disease below the age of two years is a novel finding which may imply immense public health implications. Moreover, we were unable to ascertain household contact of most of the TST-positive and clinical PTB cases in our study. This observation indicates that a considerable number of undiagnosed adult TB cases may be still available in the community. Contact tracing system is not well conducted in developing countries with a high burden of tuberculosis cases. Thus, our finding underscores the importance to screen the undernourished children for latent pulmonary tuberculosis with or without smear-positive adult household contact who may be benefited from INH prophylaxis, already recommeneded by the WHO. We have revised our conclusion on the basis of your recommendation. 

Now, we also calculated the sample size using the formula of a descriptive study: n = Z² p(1-p)/d² and found 163 participants is sufficient for this type of screening program [where, n = required sample size, Z = standard normal variant value for 95% confidence level (1.96), p = expected proportion of the condition (12.4% according to a tuberculin survey conducted in Bangladesh), and d = precision (0.05)]. However, we screened 243 children and we believe the available sample will allow us to perform a case-control analysis between TST-positive and TST-negative participants, nested in the main BEED study. We have incorporated this sample size calculation in the text on page 4.

---

## [Editor Report · Decision Letter 1]

22 Oct 2019

Impact of negative tuberculin skin test on growth among disadvantaged Bangladeshi children

PONE-D-19-21500R1

Dear Dr. Gaffar,

We are pleased to inform you that your manuscript has been judged scientifically suitable for publication and will be formally accepted for publication once it complies with all outstanding technical requirements.

With kind regards,

HASNAIN SEYED EHTESHAM

Academic Editor

PLOS ONE

Additional Editor Comments (optional):

This manuscript describes the study on the impact of the TST positivity on children’s growth in response to community based intervention. This is one of the first study of its kind and has value in terms of management of tuberculosis. The manuscript has been comprehensively revised based on the suggestions of the Reviewers. They have highlighted the positive aspects of the manuscript and also calculated the sample size which enable appropriate conclusions to be drawn. Other queries have also been addressed.
---

## [Editor Report · Acceptance letter]

28 Oct 2019

PONE-D-19-21500R1 

Impact of negative tuberculin skin test on growth among disadvantaged Bangladeshi children 

Dear Dr. Gaffar:

I am pleased to inform you that your manuscript has been deemed suitable for publication in PLOS ONE. Congratulations! Your manuscript is now with our production department. 

With kind regards,

on behalf of

Prof HASNAIN SEYED EHTESHAM 

Academic Editor

PLOS ONE